# Myeloid-Derived Suppressor Cells: New Insights into the Pathogenesis and Therapy of MDS

**DOI:** 10.3390/jcm11164908

**Published:** 2022-08-21

**Authors:** Maria Velegraki, Andrew Stiff, Helen A. Papadaki, Zihai Li

**Affiliations:** 1Pelotonia Institute for Immuno-Oncology, The Ohio State University Comprehensive Cancer Center-James, Columbus, OH 43210, USA; 2Division of Hematology, Department of Internal Medicine, The Ohio State University, Columbus, OH 43210, USA; 3Division of Medical Oncology, Department of Internal Medicine, The Ohio State University, Columbus, OH 43210, USA; 4Department of Hematology, University Hospital of Heraklion, 71500 Heraklion, Greece

**Keywords:** myeloid-derived suppressor cells, myelodysplastic syndromes, bone marrow failure, immune dysregulation

## Abstract

Myelodysplastic syndromes (MDS) are hematopoietic malignancies characterized by the clonal expansion of hematopoietic stem cells, bone marrow failure manifested by cytopenias, and increased risk for evolving to acute myeloid leukemia. Despite the fact that the acquisition of somatic mutations is considered key for the initiation of the disease, the bone marrow microenvironment also plays significant roles in MDS by providing the right niche and even shaping the malignant clone. Aberrant immune responses are frequent in MDS and are implicated in many aspects of MDS pathogenesis. Recently, myeloid-derived suppressor cells (MDSCs) have gained attention for their possible implication in the immune dysregulation associated with MDS. Here, we summarize the key findings regarding the expansion of MDSCs in MDS, their role in MDS pathogenesis and immune dysregulation, as well their potential as a new therapeutic target for MDS.

## 1. Introduction

### 1.1. Myeloid-Derived Suppressor Cells (MDSCs)

Beginning in the early 2000s, increasing reports describing tumor-associated myeloid cell populations with immune suppressive function that were distinct from macrophages began to appear in the literature [1,2,3,4]. Over the following decades, this initial trickle of findings has grown into an entire ocean of literature dedicated to understanding the development, function, and clinical implications of what are now called myeloid-derived suppressor cells [5,6]. While originally described as a heterogenous group of immature myeloid cells, MDSCs are now defined as consisting of two major populations, polymorphonuclear MDSCs (PMN-MDSCs) and monocytic MDSCs (M-MDSCs) [6]. Immunophenotypic identification of these cells has been formalized since their initial identification [7]. In humans, PMN-MDSCs are identified as CD11b^+^/CD15^+^/CD14^-^/CD66b^+^, while M-MDSCs are CD14^+^/CD15^-^/HLA-DR^-/lo^. In mice, PMN-MDSC are defined as CD11b^+^/Ly6G^hi^/Ly6C^lo^ and M-MDSC are CD11b^+^/Ly6G^-^/Ly6C^hi^ cells [8]. Unfortunately, in mice, these markers fully overlap with normal neutrophils and monocytes, requiring functional validation of the MDSC immune suppressive function. More recently, single-cell transcriptomic approaches have proven to be able to differentiate MDSCs from other myeloid lineage cells based on a unique gene expression signature [9]. These studies have led to even further diversification of the MDSC population, with the identification of early-stage MDSC (eMDSC) that lack definitive granulocytic and monocytic markers, and which consist of early myeloid progenitor cells [6,8,10]. In addition, there is increasing evidence for the ability of tumors to hijack hematopoiesis to favor the development of immune-suppressive myeloid cells. A population of monocytic lineage cells that are capable of differentiating into PMN-MDSCs has been identified, and this population was significantly expanded in the setting of cancer [11]. Furthermore, Long et al. recently showed that, in the presence of cancer, erythropoiesis is altered to favor the development of myeloid cells capable of inhibiting T cell responses over normal erythroid cells, resulting in further immune suppression and anemia [12]. These findings open exciting new lines of investigation to better understand how cancer affects hematopoiesis, as well as how to target this process for therapeutic benefit.

Currently, the development of MDSCs in the setting of cancer is thought to occur by a two-step process [13]. The first step involves the expansion of the myeloid cell compartment driven by chronic exposure to inflammatory cytokines and myeloid growth factors such as interleukin (IL)-6, IL-1β, granulocyte–macrophage colony-stimulating factors (GM-CSF), and S100A8/9, amongst others. The second step occurs within the tumor microenvironment (TME) and results in the pathological activation of MDSCs and the acquisition of the immune-suppressive function. This process is mediated by the activation of the signal transducer and activator of transcription (STAT) pathways, ER stress, Nuclear factor-κB (NF-κB), oxidized lipids, and prostaglandin-E2 (PGE2), as well as many others [6,14]. The PMN-MDSCs and M-MDSCs have been shown to inhibit both innate and adaptive immune responses by a variety of mechanisms, and in many cases these mechanisms are shared between both subsets [14,15]. For example, both PMN and M-MDSC are known to express high levels of arginase, resulting in the depletion of L-arginine from the microenvironment, thereby resulting in impaired T cell responses [16,17,18,19]. However, there are also important differences between these subsets. PMN-MDSC are known to rely more on the production of reactive oxygen species and reactive nitrogen species such as peroxynitrite [20,21,22,23]. In addition, PMN-MDSC have been shown to produce larger amounts of PGE2 via the increased expression of fatty acid transporter 2 [24]. In contrast, M-MDSC are known to express inducible nitric oxide synthase (iNOS) and cytokines such as transforming growth factor (TGF)-β and (IL)-10 [25,26,27]. Furthermore, PMN and M-MDSC are differentially recruited to the TME. PMN-MDSC are recruited predominantly by CXC chemokines, including CXCL1, CXCL5, and CXCL8, whereas M-MDSC are recruited by the C-C motif chemokine ligand (CCL)2 and CCL5 [28]. Finally, PMN and M-MDSC also have different lifespans and developmental potential within the TME. Whereas PMN-MDSC are short lived and die quickly after entering the TME, M-MDSC can differentiate into tumor-associated macrophages (TAMs) and persist within the TME [29,30].

There is an increasing number of studies on the potential involvement on MDSCs in many aspects of hematologic diseases that have been recently reviewed elsewhere [31,32]. Here, we summarize the current knowledge on the possible pathogenetic roles of MDSCs in myelodysplastic syndromes (MDS). Although it is still an evolving area of research, clearly not as well trodden as the solid tumor field or other hematological malignancies, there is much evidence and even more speculation that MDSCs might be involved in the entire spectrum of MDS—from the emergence of malignant clones to clonal evolution, to immune evasion and treatment resistance.

### 1.2. MDS Molecular Pathogenesis

MDS are hematopoietic stem cell disorders characterized by myeloid lineage dysplasia, bone marrow (BM) failure, and a propensity to transform to acute myeloid leukemia (AML) [33,34]. It appears that the development and progression of MDS is dependent upon the acquisition of specific somatic mutations and the development of a dysregulated immune response. Our understanding of how these two processes contribute the development of MDS and interact with each other has improved significantly over the past decade [35,36].

Initial investigation of the genetic drivers of MDS focused on cytogenetic abnormalities [37,38]. These studies showed that around 50% of MDS patients had cytogenetic abnormalities with unbalanced changes resulting in losses or gains of chromosome regions, with -7/del(7q) and -5/del(5q) being the most common. The biological significance of these cytogenetic alterations was demonstrated by their prognostic significance in the Revised International Prognostic Scoring System (IPSS-R) for MDS [39]. Building on these findings, next generation sequencing studies have identified recurrent somatic mutations in MDS patients. These studies showed that MDS has a relatively low mutation burden, but there were a number of genes with recurrent mutations, including those involved in DNA methylation (*DNMT3A, TET2, IDH1/2)*, RNA splicing (*SF3B1, SRSF2, U2AF1*), and histone modification (*EZH2* and *ASXL1*) [36,40,41,42,43,44,45,46,47,48]. While not all the functional consequences of these mutations are known, it appears these mutations serve as drivers for MDS by dysregulating the expression or function of key genes involved in hematopoiesis, such as GATA1, KLF1, and HOXA9 [49,50,51,52,53,54].

In addition to directly dysregulating hematopoiesis, some MDS-associated cytogenetic abnormalities and somatic mutations are associated with chronic inflammation and the dysregulation of innate immune system signaling pathways. Inflammatory factors within the BM are known to impact HSC function, self-renewal, lineage differentiation choices, and progenitor cell maturation. The inflammatory BM milieu is known to be implicated in a wide spectrum of states, ranging from age related anemia to BM failure syndromes and MDS development, which is more likely in the case of concurrent genetic lesions [35]. The interplay between the inflammatory BM microenvironment and acquired driver mutations is considered crucial for the development of MDS. Very interestingly, TGFβ1 was recently shown to shape the outcome of coexisting inflammatory stimuli, favoring BM failure in the absence of genetic factors [55]. This adds another layer of comprehension of the diversity of the inflammation-associated outcomes in the BM and the clinical presentation of the BM failure syndromes, as well as the understanding of the multifaced roles of TGFβ1 in hematopoiesis [56]. In MDS, the importance of dysregulated innate immune system signaling and inflammation is evidenced by several studies showing that the innate immune receptor Toll-like receptor (TLR)-4, and many proteins involved in its signaling pathway, are overexpressed in MDS [57,58,59,60,61]. Overexpression of TLR4 makes MDS cells highly responsive to damage-associated molecular patterns (DAMPs) such as the S100A8/A9 heterodimer. The S100A8/A9 ligation of TLR4 activates signaling pathways such as NF-κB and mitogen-activated protein kinase, resulting in the production of inflammatory cytokines such as IL-6, tumor necrosis factor (TNF)α, and IL-1β, amongst others [62,63]. Molecular studies investigating the functional consequences of the loss of del(5q) genes further emphasize the importance of TLR4 signaling in MDS. The *MIR146A* and TRAF-interacting protein with the forkhead-associated domain B (*TIFAB)* are two 5q genes that are lost in the majority of MDS patients with del(5q) [64,65]. Decreased levels of miR-146a and TIFAB lead to increased tumor necrosis factor receptor-associated factor 6 (TRAF6) protein levels, a key protein in the TLR4 signaling pathway, causing enhanced TLR4 activation and ineffective hematopoiesis [66,67,68]. TLR4 stimulation also leads to activation of the NLRP3 inflammasome, resulting in increased IL-1β production and cell death via pyroptosis, contributing to cytopenias [69]. Interestingly, *TET2* mutations that commonly occur in MDS are associated with chronic inflammation. For example, TET2 appears to negatively regulate IL-6 via the recruitment of HDAC2, suggesting that MDS-related inactivating *TET2* mutations could lead to increased IL-6 expression [70]. Furthermore, the loss of TET2 has also been associated with the increased activation of the NLRP3 inflammasome and IL-1β levels [71,72]. This chronic inflammatory microenvironment serves as a natural link to the promotion of MDSCs. S100A8/A9, while being critical to the development of MDS, as detailed above, is also well known to promote the development of MDSCs [73]. Furthermore, many of cytokines that are expressed downstream of TLR4 activation also promote MDSC expansion and activation, such as IL-1β, IL-6, and TNFα [74,75,76,77]. This scenario creates the potential for a feed-forward loop to develop between MDS and MDSC generation, leading to the development of a highly immune-suppressed microenvironment.

## 2. Expansion and Significance of MDSCs Derived from Nonmalignant Clones in Patients with MDS

It was first documented by Chen et al., and later also reported by other groups, that MDSCs accumulate in the BM of patients with MDS, as indicated by the increased proportion of the Lin^-^/CD33^+^/CD11b^+^/DR^−^ population in MDS patient peripheral blood (PB) and BM samples [78,79,80,81,82]. An analogous population has been identified by the unsupervised clustering of multidimensional mass cytometry data in patients with MDS [83]. The expansion of MDSCs has been mainly observed in the patients classified as high risk [80,81,82,84]. Although it is still an open question under investigation, there is more evidence today, mainly from preclinical models [78,85], that the presence of MDSCs in the BM per se can induce myelodysplastic changes and, therefore, the accumulation of MDSCs in the BM of MDS patients could be of pathogenetic significance contributing to the development and clinical course of the disease, as is discussed in detail in the next section.

Cytogenetic studies of MDSCs, isolated from the BM of patients with MDS-related chromosomal abnormalities, has revealed that the accumulating MDSCs do not bear the genetic lesions of the MDS clone, indicating that they do not derive from the malignant hematopoietic stem cells (HSCs) [78]. This has been further confirmed by molecular studies which did not identify any of the common MDS-associated mutations in the purified MDSCs from MDS patients.

Despite their nonmalignant origin, notable qualitative differences have been reported between MDSCs isolated from MDS patients and those from healthy donors. Increased levels of the CD33 expression are among the most important MDS-related MDSC features that have been reported [78], with promising translational potential. Specifically, Chen et al. showed that MDSCs derived from MDS patients have unusually increased levels of CD33, and this is considered a key factor for their aberrant expansion, since they showed that CD33 can directly interact with S100A9 [78]. This previously unrecognized ligand–receptor relationship between S100A9 and CD33 provides an explanation for earlier observations that S100A9 can induce the expansion of MDSCs [73]. Moreover, S100A9 levels were found to be elevated in the BM plasma of MDS patients, in line with the hypothesis that CD33 mediates MDSC expansion in MDS patients. They also showed that the clonogenic capacity of HSCs significantly improves by knocking down CD33 in the cocultured MDSCs, and vice versa, CD33 could repress myelopoiesis when it was overexpressed in healthy donor-derived MDSCs, suggesting that the downstream activated pathways may play a role in the BM failure of the MDS patients. This important role of CD33 has been utilized in the clinic by developing CD33-targeting therapies to improve the MDS-related cytopenias, as will be discussed later.

Although their actual pathogenetic role has not been fully elucidated, other phenotypical changes that have been reported and linked with the accumulation of MDSCs in the BM of MDS patients, include altered chemokines and chemokine receptors CXCR4, CX3CR1 [80], CCR2, and CCL2 [81]. Altered programmed cell death protein 1 (PD1) expression in MDS patients with TP53 mutations has also been reported, though the significance of this finding has not been explored so far [86].

## 3. MDSCs as Inducers of Myelodysplasia

The first genetic evidence for the pathogenetic significance of MDSCs in MDS came from a study where S100A9 transgenic (Tg) mice were used as a model of the progressive expansion of MDSCs in the PB, spleen, and the BM [78], utilizing the knowledge that MDSCs can be induced by S100 proteins [73]. Interestingly, the observed numerical changes of the Gr1^+^CD11b^+^ MDSCs in the BM of the Tg mice over time were accompanied by a gradual decrease of hemoglobin levels and a drop of neutrophil and platelet counts. Synchronously, the BM of the mice was markedly hypercellular and showed several dysplastic morphological changes that resembled those of MDS in humans. Collectively, these data suggested that the expansion of MDSCs in the BM of mice can induce anemia associated with myelodysplastic changes in the BM. The contribution of BM MDSCs to this phenotype was further supported when lineage-negative enriched-HSPC from S100A9Tg mice were adoptively transferred to lethally irradiated WT recipients; the observed increase in the numbers of MDSCs in the BM of the latter after engraftment was accompanied by low Hb levels in a similar way to the aged transgenic mice. When both WT and S100A9Tg HSPC were transplanted as a mix, the recipient mice were initially healthy, but gradually developed late onset anemia, suggesting that the presence of the expanding MDSC population deriving from the S100A9Tg component could still interfere with hematopoiesis of the WT donor cells, demonstrating that MDSCs can bear an actual pathogenetic role in the anemia of MDS by suppressing normal hematopoiesis.

The same group added another significant layer of evidence for the deleterious effect of MDSCs on hematopoiesis, using all-trans-retinoic acid (ATRA) to induce the differentiation of the accumulated MDSCs in the S100A9Tg mice [78]. As expected, treatment of the S100A9Tg mice with ATRA resulted in a significant drop of the Gr1^+^CD11b^+^ MDSC numbers. This reduction of MDSCs was followed by an increase in all hematologic parameters at levels similar to that of the WT mice, suggesting that the MDS-like phenotype of the S100A9Tg mice is driven by the presence of MDSCs in the BM, and that eliminating this population is sufficient to rescue the mice from developing anemia in this model.

In an independent study, Mei et al. generated mice with the dual deletion of miR-146a and mDia1 to explore the pathophysiologic role of genes on chromosome 5q in the BM failure of del(5q) MDS [85]. These double knockout (KO) mice developed anemia over time, along with the morphological features of red blood cell lineage dysplasia in their BM. Interestingly, there was an accumulation of Gr1^+^Mac1^+^ immunosuppressive MDSCs in their BM and PB, which seemed to be the main cellular source of TNF-a and IL-6 in this model, and was considered responsible for the disturbed erythropoiesis that was observed. Administration of ATRA, which induced the maturation of MDSCs, again rescued the anemia and BM failure both in a therapeutic and a prophylactic mode, indicating that there is a pathogenetic association between the presence of the MDSCs in the BM and ineffective hematopoiesis. The same group later reported that the double miR-146a and mDia1 KO mice developed leukemia after their first year of life, mimicking the progression to AML seen in some patients with MDS and, interestingly, this trajectory changed in the absence of IL-6, highlighting the significance of IL-6 production in the above model [87]. Whether the rescuing effect of the IL-6 deletion is mediated through the abrogation of the MDSC accumulation in this model remains to be clarified. Yet, the IL-6-induced activation of the STAT3 pathway has been proposed as a possible mechanism of MDSC accumulation in MDS [81]. This could be a likely scenario in the IL-6-rich BM microenvironment of MDS patients, since MDSCs are known to escape necroptosis after DNA methylation following IL-6-induced STAT3 signaling in the context of cancer [88].

Diving further into the possible mechanisms of the MDSC-mediated myelodysplasia, P. Cheng et al. reported that elevated levels of S100A9 in the BM of MDS patients may account for the observed increased expression of PD1 and PD-L1 on the HSCs and BM MDSCs, respectively [89]. The induction of the PD-1/PD-L1 axis subsequently induces caspase-3 and eventually cell death both in MDS patients and the S100A9 Tg mice, contributing to ineffective hematopoiesis. To support this hypothesis, they did several in vitro experiments, exposing normal human CD34^+^ HSCs or CD33^+^CD14^+^ MDSCs to either recombinant S100A9 or plasma from MDS patients, which both activated the PD-L1/PD-1/caspase-3 axis. Interestingly, the presence of anti-PD-1 or anti-PD-L1 blocking antibodies significantly improved the clonogenic capacity of MDS BM cells, further indicating that the S100A9-induced PD-1/PD-L1 pathway activation may directly contribute to the BM failure of MDS patients. Most importantly, they showed in vivo that the treatment of S100A9 Tg mice with the anti-PD1 inhibitor significantly improves the colony-forming capacity of the BM cells and increases most of the hematological parameters, further reinforcing the role of this pathway in the ineffective hematopoiesis of MDS.

All the above studies have provided important insights into the pathophysiological significance of the expanding MDSCs in the BM and their potential role in the ineffective hematopoiesis, even though many of these observations are based on transgenic murine models and, therefore, the common limitations of them, such as the induction of nonphysiological changes, should be taken into consideration when evaluating these findings. However, there are still many open questions regarding the mechanisms and regulation of their expansion, as well as their actual pathogenetic contribution, not only to disease progression, but also to the disease initiation, which future studies are expected to address. It is not known so far, for example, if MDS patients bare polymorphisms in MDSC regulating genes, which could provide a direct link between genetic predisposition, MDSCs, and MDS development.

## 4. MDSCs and Immune Dysregulation in the MDS

Along with the direct impact of MDSCs on the HSCs, the presence of MDSCs in the BM of MDS patients is expected to have further impact on the BM microenvironment and the disease progression stemming from their known robust immunosuppressive properties. Although it is predicted that the accumulating MDSCs would inevitably alter the local immune dynamics in the MDS BM, in-depth studies that illustrate the exact contribution of MDS MDSCs to the associated immune dysregulation are yet to be done. Here, we specify some of the recent reports that have sought to provide insights on the potential involvement of MDS MDSCs to the associated aberrant immune responses, which is a well-described phenomenon in MDS, as described above [35,90,91].

Chen et al. showed that Lin^-^CD33^+^ MDSCs purified from the BM of MDS patients secrete more IL-10 and TGFβ1 compared to MDSCs derived from healthy donors [78]. High CD33 expression on patients’ MDSCs seems to account, at least in part, for the increased cytokine production, since knocking down CD33 from the MDS-derived MDSCs results in lower levels of the secreted cytokines. In accordance with this, the artificial overexpression of CD33 in normal BM cells was accompanied by the increased production of both IL-10 and TGFβ1, especially after treatment with rhS100A9, in a CD33-dependent fashion. Therefore, the S100A9–CD33 axis is an important driver of suppressive cytokine production by MDSCs in MDS BM. In line with these data, it has also been reported that the levels of IL-10 and TGFβ1 produced by Lin^−^/CD33^+^/ DR^−^ MDSCs are increased in high-risk MDS patients, implying their possible prognostic value in patients with MDS [92]. Moreover, MDSCs could be themselves cellular sources of proinflammatory cytokines such as IL-1β [84], contributing to the inflammatory milieu, a well characterized feature of MDS [93].

One of the questions that seems to have been successfully addressed so far without major controversies in the literature is whether this population with the phenotypical characteristics of MDSCs found in the BM of MDS patients can still exert their suppressive functions over conventional T cells. It has been shown that CD14^+^/DR^−^ MDSCs isolated from MDS patients are capable of suppressing the proliferation of both allogeneic and autologous CD4 cells in vitro, although they were not directly compared with the suppressive capacity of healthy donor-derived MDSCs [80]. It has been also reported that, after coculture with Lin^-^/CD33^+^/DR^-^ MDSCs purified from MDS patients, CD8+ T cells show decreased proliferation and lower expression of perforin and granzyme B [94]. As a possible mechanism, the authors proposed that increased Galectin 9 (Gal9) production by the MDS-derived MDSCs initiates the T cell exhaustion program and accounts for the observed phenotypically defined dysfunctional state of the CD8+ T cells after their coculture with the MDS MDSCs, although this was not directly proven in this study. It was shown, however, that in the presence of TIM3/Gal9 inhibitors, the suppressive effect of the MDSCs on CD8+ cells was abrogated, indicating that the Gal9-TIM3 axis may play a role in the MDS MDSC-mediated T cell suppression. The possibility of MDSCs inducing T cell exhaustion in MDS has been also raised by another group, with the secretion of CEACAM1 being one of the factors proposed to mediate this action [84]. In another study, it has been proposed that the STAT3/Arg1 axis is involved in MDSCs-mediated immunosuppression in MDS, since again the pharmacological inhibition of STAT3 was able to improve the MDSC-induced changes of effector molecules on CD8+ T cells in vitro [81]. Collectively, it seems that not only do MDSCs preserve their suppressive function, but they are also aberrantly activated in the MDS BM, and this could be a contributing factor to the defects of the of the innate and adaptive immunity in MDS.

## 5. MDSCs as Therapeutic Targets in MDS

Given the potential role of MDSCs in the pathogenesis and progression of MDS, there has been increasing interest in developing MDSC-targeting modalities to improve hematopoiesis in MDS. Such approaches are expected to reform the BM in MDS by altering the immune environment, as well as inhibit the direct interactions of MDSCs with the malignant clone. As we discuss below, the encouraging preclinical data have not always been translated into clinical benefit in MDS patients, and the ambition of transferring the new knowledge of the pathogenetic aspects of the disease to efficient therapeutic regiment designing remains a big challenge.

Based on the findings of their previous study, Eksioglou et al. examined the potential of targeting MDSCs to restore hematopoiesis in MDS. They tested the BI 836858 humanized anti-CD33Ab, with an engineered IgG heavy chain, to target the CD33^high^ MDSCs [95]. They used this antibody to treat bone marrow mononuclear cells (BMMCs) isolated from MDS patients and showed that it significantly increases the clonogenic capacity of the MDS-derived samples to normal levels. Mechanistically, they showed that BI 836858 induces Natural Killer (NK) cell-mediated antibody-dependent cellular cytotoxicity (ADCC) of MDSCs, as was indicated mainly by the significant reduction of the Lineage^−^HLA^-^DR^−^CD33^+^ cell fraction of MDS-derived BMMCs after their ex vivo treatment with BI 836858. Along with the cytotoxic effect, they also found that BI 836858 can block the CD33 downstream pathway, as demonstrated by the reduction of IL10 and ROS production by BM cells after BI 836858 treatment, and they further showed that BI 836858 can reduce the genomic instability induced by S100A9.

Due to the promising findings of these ex vivo studies, BI 836858 was later tested in a phase I/II, dose escalation randomized trial in patients with low/intermediate-1 risk MDS (NCT02240706). Unfortunately, the trial was prematurely terminated during the dosing escalation phase before reaching the planned maximum dose, as there were no observed hematological responses in other than one patient, who showed a noteworthy increase in their hemoglobin levels [96]. The main reason for the failure of this trial seems to be that BI 836858 failed to reduce the numbers of MDSCs, despite an ostensible reduction in CD33 expression that was attributed to internalization of the antigen and not to real elimination of the MDSC population in the BM of patients. This could be due to the pre-existing NK cell dysfunction that was not improved after the treatment with BI 836858, as was expected according to the preclinical data. This lack of induced cytotoxicity, and the subsequent failure to successfully reduce the numbers of MDSCs in MDS patients, makes this study unsuitable for evaluating the relevance of targeting MDSCs to improve hematopoiesis in MDS, since the aim of abolishing the MDSCs was not achieved. Eliminating MDSCs in MDS could still be considered as an enticing concept with a promising therapeutic potential; however, it might be challenging to achieve with a single, ADCC-based, agent in the multidysfunctional immune environment of MDS BM [97,98,99].

For the reasons discussed above, bispecific engagers are expected to confer better clinical efficacy. Using a CD16xCD33 bispecific killer cell engager (BiKE) that was originally developed to target the CD33^+^ clone in AML [100], Gleason et al. showed that, despite the low expression of CD16 in MDS patients, MDS-derived CD16^+^ NK cells can be triggered ex vivo to effectively lyse the CD33 expressing targets, including allogeneic in vitro-generated MDSCs, overcoming the common suppressive effect of the latter on the NK cells [101]. The same concept has been tested with a TriKE construct, generated by adding an IL-15 linker to the CD16xCD33 BiKE, to improve longitudinal NK resistance to the MDSC inhibitory effects [102]. The expectation of these modalities is that they will improve NK cell function, leading to cytotoxicity against CD33 expressing clones and MDSCs. In their ex vivo studies, they showed that the 161533 TriKE can improve the NK-cell-mediated cytotoxicity even in the presence of MDSCs, and in accordance with that, they showed that it has a substantial effect on NK cell proliferation and NK cell responses within the MDS-derived PBMCs, overcoming the fact that there was a profoundly reduced NK cell component within the initial MDS samples tested [102]. The GTB-3550 TriKE is currently being tested in high-risk MDS and refractory AML patients in an ongoing phase I/II clinical trial (NCT03214666).

In a similar mode, the CD33/CD3 bispecific T cell engager AMV564 9, which has already been shown to have cytotoxic effect on CD33+AML cell lines, has been reported to effectively reduce the number of MDS-derived BM MDSCs ex vivo and improve the clonogenic capacity of MDS BMMCs [103]. It is currently being tested in a phase 1 clinical trial in patients with intermediate-2 or high-risk MDS (NCT03516591), and the results of this study will provide more information on the efficacy of this agent in depleting the CD33-expressing malignant clone and MDSCs, as well as the clinical benefit from it.

Another molecule which is expressed by both the malignant clone and the MDSC population in patients with MDS and AML is the IL-3 receptor α-chain CD123 [104,105,106]. This dual expression makes CD123 another promising target to concurrently eliminate the blasts and the MDSC component in the BM of MDS patients. Several anti-CD123 agents are being currently tested in AML and high-risk MDS, and are being reviewed elsewhere [105]. APVO436 CD3xCD123 bispecific antibody has shown promising activity in a small cohort of previously treated MDS patients [107]; however, no data have become available regarding the effectiveness of this agent in eliminating the MDSC population and its possible association with the clinical outcome. Interestingly, chimeric antigen receptor (CAR) T cells targeting CD123 ± CD33 are also being tested under phase 1 clinical trials in patients with high-risk hematologic malignancies including MDS (NCT04156256, NCT03795779), which is currently recruiting.

Apart from directly depleting them, there are several other ways to target MDSCs for therapeutic purposes, such as inducing their maturation, blocking their proliferation and migration, impeding their metabolism, and inhibiting their suppressive function. There are several modalities targeting these pathways that are currently being tested in patients with cancer, with promising effects [108]. The inhibitor of Indoleamine 2,3-Dioxygenase (IDO) Enzyme INCB024360 was recently tested in a phase 2 clinical trial in patients with high-risk MDS who had failed to respond to hypomethylating agents [109]. A small proportion of the patients showed a slight reduction in the number of their MDSCs and improvement of their colony-formation capacity in vitro, which was not translated to a clear clinical benefit since, in this very poor prognostic patient group, the best response that was achieved was stable disease.

Treatment approaches of targeting MDSCs constitute a new paradigm of therapeutics in solid tumors and hematologic malignancies. Although they have not been as extensively tested in MDS as in other disease entities, they are considerably relevant in MDS given the well-known association of immune dysregulation and the emerging multifaced roles of MDSCs, immune and nonimmune, in the development and progression of the disease. It is expected that the basic and translational research on this field will be broadly expanding in the future, providing new possibilities and options for the treatment of MDS patients, as is the case with other developing immune targets in MDS [91].

## 6. Future Perspectives

As detailed above, the growing body of literature investigating the potential involvement of MDSCs in MDS points towards MDSCs having important roles in many aspects of MDS, like the development of the malignant clones, the immune dysregulation, clonal expansion, and leukemic transformation (Figure 1). Further work is needed to better understand both how MDS clones induce the expansion of MDSCs, as well as further clarify the role and potential mechanisms by which MDSCs can induce BM failure and dysplasia.

As discussed earlier, a key step in the initiation of MDS is the acquisition of cytogenetic abnormalities and/or mutations in MDS-associated genes, such as those involved in DNA methylation, RNA splicing, and histone modifications. While some functional studies, such as those investigating the consequences of loss of del(5q) genes, have linked genetic changes to activation of dysregulated innate immune system function and inflammation, whether and how this occurs with other MDS-associated genetic changes is not clear. An interesting area to explore in the future could be the mutations associated with clonal hematopoiesis of indeterminate potential (CHIP). Many CHIP-associated mutations overlap with those found in MDS [110]. Furthermore, there is evidence that CHIP mutations may drive chronic inflammation, which could serve as a link to the expansion of MDSCs and the potential development of MDS [111,112]. However, clearly there is not an expansion of MDSCs in all CHIP patients, and not all CHIP patients develop MDS, so there must be additional important factors that are not yet identified. One concept that could potentially link the inflammation from CHIP mutations with the expansion of MDSCs is trained immunity (TI). TI is an epigenetic and metabolic mechanism whereby the stimulation of innate immune cells with PAMP/DAMP molecules produces a more rapid and robust inflammatory response secondary to stimulation with the same or similar molecules [113]. Recent studies have shown that TI functions at the level of HSCs and myeloid progenitors in the BM, resulting in the alteration of myeloid cell differentiation and function of downstream terminally differentiated populations [114,115]. Mechanistically, this process relies in part on the increased production of IL-1β within the BM, which stimulates the cycling of long-term HSC and the myeloid skewing of hematopoiesis [115]. Currently, whether CHIP or MDS clones can induce a TI response in normal HSC is unknown.

More work is also needed to understand how MDSCs, once generated, interact with HSC and early progenitor populations to alter their developmental potential. In the past few years, there has been increasing evidence that soluble factors derived from tumor cells exert significant effects on the early stages of hematopoiesis, leading to the significant modulation of the BM function. In 2015, Casbon et al. demonstrated that breast-cancer-derived G-CSF induced the substantial remodeling of hematopoiesis, which resulted in myeloid cell expansion and concurrent anemia. More recently, Long et al. built on this work by showing that tumor-derived GM-CSF reprograms the developmental potential of erythroid precursor cells, resulting in their acquisition of a myeloid phenotype and immune suppressive function [12]. AML blast-derived IL-6 was recently found to inhibit erythropoiesis at the erythroblast phase, resulting in severe anemia independent of the BM blast percent [116]. Given that MDSCs appear to be a rich source of inflammatory factors within the BM, it will be important to better characterize specifically what cytokines MDS-associated MDSCs produce, as well as what effect these molecules have on HSC function, lineage differentiation choices, and possible lineage plasticity. Beyond soluble factor production it also important to explore what, if any, effect other known mechanisms of MDSC immune suppression have on hematopoiesis. For example, as discussed above, PMN-MDSCs produce large amounts of reactive oxygen and nitrogen species. These molecules are highly reactive and have been shown to alter protein structure, resulting in the loss of function and the inhibition of signaling pathways. In MDS, this could lead to the dysfunction of proteins and pathways that are critical to differentiation, leading to arrested maturation and cell death. The role of MDSC-derived reactive oxygen and nitrogen species in MDS is currently completely unknown. A better understanding of all these processes in MDS has the potential to yield new therapeutic targets for MDS.

A thorough characterization and deep understanding of the interactions between the MDSCs and the malignant clone in the MDS BM, and the pathophysiologic significance of it, entails the development of animal models that successfully recapitulate every aspect of MDS pathogenesis. So far, the most convincing evidence of the possible roles of MDSCs in inducing myelodysplasia comes from the two murine models described above. Hopefully, in the future, new mouse strains that model better the disease will serve as better platforms to study the interplay between MDSCs and MDS clones. Human studies shall be expanded beyond the genomic effort to also include deeper studies of the BM microenvironment, using other state-of-the-art technology such as genetic tracing, imaging, and multicomics. Eventually, the better understanding of the disease pathogenesis will provide the platform for novel therapeutic development. These efforts need to be systematic, mechanism-driven, and carefully controlled.

## Figures and Tables

**Figure 1 jcm-11-04908-f001:**
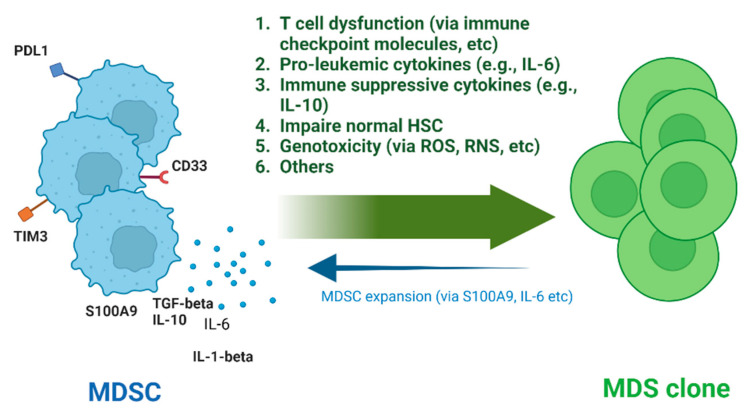
Pathogenetic roles of MDSCs in MDS. HSC, Hematopoietic Stem Cell; ROS, Reactive Oxygen Species; RNS, Reactive Nitrogen Species (Created with BioRender.com.)

## Data Availability

Not applicable.

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
