# Peer review of "Myeloid-Derived Suppressor Cells: New Insights into the Pathogenesis and Therapy of MDS"

_jcm, 2022, doi:10.3390/jcm11164908_

Round 1

Reviewer 1 Report

Velegraki et al. present a review of myeloid derived suppressor cells involved in myelodysplastic syndromes. The topic is interesting and deserves attention, however some improvements could be made. The description of the MDSC could be a little more detailed, for example by adding the suppression mechanisms used to perform the main function. The MDS disease is described minimally, a mention of its classification and some other characteristic would be of help to the reader. Paragraph three could be better explained, it is confusing and unclear than the rest of the text. More attention should be paid to the nomenclature of proteins and genes, the use of acronyms the first time they are mentioned in the text, and the use of the same wording when talking about a model (eg line 204-209).

Line 34: It would be preferable to insert a more recent reference. Suggestion: insert a table with the MDSC markers for each subset, at least in humans. Maybe a list of markers and positivity and negativity in each subset would allow a better comparison.

Line 37-39: The definition of human MDSCs is more problematic, as several cell subsets have been described (Ref 7), however the cell identity complexity can be simplified into three main groups: M-MDSCs, PMN-MDSCs and early-MDSCs. In addition, I suggest inserting the reference PMID: 27381735 where the characteristics necessary for the attribution of the name MDSC are defined.

Line 59: Suggestion: It would be better to use the full name of proteins the first time they are introduced into the text (eg interleukin (IL)-6). This tip applies to all text.

Line 62: Acronym STAT

Line 68: Has it been shown that the deletion of some transcriptional factors in the myeloid lineage leads to an abrogation of the immunoregulatory properties of MDSCs? If so, by what mechanisms?

Line 74: Interleukin

Line 77: Acronym CCL

Line 110: Can the authors give example of key genes involved in hematopoiesis?

Line 116: Acronym TLR

Line 137: Can the authors give example of cytokines produced as a result of the activation of TLR4?

Line 150: Can the authors add the references of preclinical studies mentioned in the main text?

Line 159-175: It is interesting to note the potential role of the protein S100A9 in the pathogenesis of MDS. However, what is known about S100A8? Since the two proteins function as heterodimers, has the alteration of S100A8 in MDS patients ever been evaluated?

Line 169-172: What about studies on preclinical models? In addition, it would be preferable to indicate what is co-cultured and which pathways are activated.

Line 184-193: How is the increase in MDSCs affected by S100A9? What causes this increase? In addition, it would be interesting to know whether the protein alteration in the transgenic mouse is total or conditional. In this model, is the role of the S100A9 due to increased expression or massive release?

Line 184-201: The text may be reorganized a bit. The concept is confused, it would be better to make it clearer.

Line 204: Can the authors explain the function performed by ATRA on S100A9? In addition, does the described impact occur on all MDSC or on a specific subset?

Line 204-209: Suggestion: It would be better to use the same name of mouse models in order to not to confuse the reader.

Line 210: Are there any studies with immunodeficient mice with BM cells obtained from MDS patients compared to healthy donors that confirm the results obtained in the preclinical model? Furthermore, has this mechanism been validated in vitro in the human context?

Line 223: Is this mechanism due to generic IL-6 or its production by specific cell types? A more detailed description would be appreciated.

Line 230-232: Are elevated levels of S100A9 related to gene or protein?

Line 235-237: Can the authors explain if is referred to human or mouse context?

Line 252: Are there any other studies on MDSC and MDS involving other cytokines or DAMPs?

Line 254-300: There are no preclinical studies on transgenic mice for CD33?

Line 317: Acronym ADCC

Line 321: Can authors explain the meaning of MB cells?

Reviewer 2 Report

Velegraki et al reviewed the role of MDSCs in the development of MDS, emphasizing the interplay between the proinflammatory environment and MDS clones. And then specifically noted MDSCs in MDS patients are from wild-type clones. The authors further discussed several key papers in the field that utilized transgenic mouse models showing MDSC may contribute to ineffective erythropoiesis in mice. Furthermore, the authors reviewed the immunomodulatory function of MDSCs on T cells in MDS and the potential of therapeutically targeting MDSCs. Overall, the review is well-written and sufficiently addressed the new findings on this topic. The reviewer has a few suggestions that may help readers to understand this topic better, as listed below.

*The authors discussed proinflammatory stress mediated by MDSCs may synergize with the existing dysregulated immune system towards the development of MDS. Of note, inflammation signals could cause bone marrow failure, or less severe, anemia which is commonly observed in aged healthy people. MDS transformation/diagnosis in clinical often requires additional cytogenic abnormalities or mutations. They could either be existing mutations (CH) or acquired. Furthermore, MDS especially lower-risk ones are heterogenous and often preserve an intact clonal architecture where mutated cells coexist with wild-type cells spanning multiple cell states. The authors should address the increased inflammatory stress in aging, more importantly, dissect by cell states how dysregulated immune signals affect human/mouse HSCPs differentiation/self-renewal, eg. HSC renewal, lineage commitment in early progenitors, ERP, and erythroblast maturation. For example, Filippi lab’s work on TGFb. Next, expand on how mutations synergize with the pro-inflammatory environment toward disease development.

*Line 277- 300. The immunomodulatory role of MDSCs on T cells can be separated as a stand-alone paragraph. Authors could expand this section by discussing immunomodulatory effects in other malignancies eg. AML or applications eg. CAR-T since the concept is well established.

*It would be helpful to make a table of literature mentioned in this review that supports the six listed effects of MDSC on MDS in figure1 and include a summary showing system (mouse or human) and sample size etc.

It would be better to list the immunophenotype of cell states when citing a paper as different groups define their cells differently. Eg. HSC

Line 221. The majority of MDS patients do not transform to AML. AML transformation should not be considered the “natural history of MDS” in human.

Authors should address murine models often present supraphysiological changes in gene expression or cytokines.
